# Suppression of migration and invasion by taraxerol in the triple-negative breast cancer cell line MDA-MB-231 via the ERK/Slug axis

**Yu-ting Xia[1,2], Yu-qin Zhang[1,2], Lu Chen[2], Liangliang Min[2], Da Huang[2], Yulu Zhang[2], Cong Li[2], Zhi-hua Li[2]***

**1** Jiangxi University of Traditional Chinese Medicine, Jiangxi, China, **2** Key Laboratory of Breast Diseases in Jiangxi Province, Third Hospital of Nanchang, Jiangxi, China

* huazhili0802@163.com

**Data Availability Statement:** We have uploaded the raw experimental data to Figshare, a stable public repository. The DOI is 10.6084/m9.figshare. 24114909.

## Abstract

As one of the triterpene extracts of Taraxacum, a traditional Chinese plant, taraxerol (TRX) exhibits antitumor activity. In this study, we evaluated the effects of TRX on the migration and invasion of MDA-MB-231 cells, analyzed the molecular mechanism through network pharmacology and molecular docking, and finally verified it by in vitro experiments. The results showed that TRX could inhibit the migration and invasion of MDA-MB-231 cells in a time- and concentration-dependent manner, while MAPK3 was the most promising target and could stably combine with TRX. In addition, the relative protein expression levels were detected by Western blot, and we observed that TRX could inhibit the migration and invasion of MDA-MB-231 cells via the ERK/Slug axis. Moreover, an ERK activator (tert-butylhydroquinone, tBHQ) partially reversed the suppressive effect of TRX on MDA-MB-231 cells. In conclusion, TRX inhibited the migration and invasion of MDA-MB-231 cells via the ERK/ Slug axis.

## Introduction

Breast cancer (BC) is a highly heterogeneous malignant tumor. Its incidence rate ranks first among female cancers. TNBC is a subtype with no expression of estrogen receptor (ER), progesterone receptor (PR) or human epidermal growth factor receptor-2 (HER-2). It has a high rate of metastasis and recurrence [1], accounting for approximately 20% of all BCs [2]. Due to negative receptor expression, TNBC is insensitive to endocrine or targeted therapy and is the subtype with the most limited treatment, the worst prognosis and the shortest survival period. Therefore, it is necessary to find new treatments and drugs for TNBC.

Meanwhile, traditional Chinese medicine (TCM) has unique antitumor advantages and tends more to the overall treatment of patients. As a traditional antipyretic and antidote medicine with a bitter taste and cold properties, Taraxacum has a great effect on detoxification, detumescence and lump dissipation and is commonly used in the treatment of breast disease. Modern pharmacological studies have shown that the extracts of Taraxacum can inhibit the occurrence and development of BC. TRX, a pentacyclic triterpene, is one of the most active ingredients and not only has anti-inflammatory [3] and antiviral effects [4] but also inhibits

**Funding:** This work was supported by Natural Science Fund in Jiangxi Province (Contract grant number: 20202BAB206046) and Jiangxi Provincial Postgraduate Innovation Special Fund Project (Contract grant number: YC2021-S500).

**Competing interests:** The authors have declared that no competing interests exist.

cell proliferation in various tumor cell lines. Previous studies have shown that TRX could inhibit cell metastasis through the Hippo and Wnt signaling pathways in gastric cancer cells [5], promote cervical cancer cell apoptosis by the mitochondrial pathway [6] and inhibit cell proliferation of bladder cancer [7] via the Akt pathway. In addition, TRX was verified to promote autophagy by suppressing mTOR phosphorylation in the BC cell line MCF-7 [8]. However, the mechanism of TRX in the treatment of TNBC is still unclear.

In this study, we evaluated the effects of TRX on the migration and invasion of MDA-MB-231 cells, analyzed the potential targets and pathways through network pharmacology and molecular docking, and finally verified them by in vitro experiments.

## Materials and methods

### Materials

MDA-MB-231 human TNBC cell line was obtained from the Key Laboratory of Breast Diseases in Jiangxi Province and cultured in high glucose DMEM with 10% fetal bovine serum (FBS). DMEM was purchased from Solarbio (Beijing, China), while FBS was purchased from Gibco (New York, USA). TRX (purity>98%, CAS:127-22-0) was purchased from Herbest Biochemical Technology Co., Ltd. (Baoji, China) and then dissolved in dimethyl sulfoxide (DMSO) from Macklin (Shanghai, China) to prepare a 20 mmol/L (mM) stock solution for storage at -40˚C. tBHQ (purity>99%, CAS:1948-33-0) was purchased from MedChemExpress (Shanghai, China) and dissolved in DMSO to prepare a 10 mM stock solution for storage at -40˚C.

Primary antibodies against epithelial-cadherin (E-cadherin, ab40772), neural-cadherin (N-cadherin, ab76011), Vimentin (ab92547), and Slug (ab85936) were purchased from Abcam (Cambridge, UK), while phospho-ERK1/2 (28733-1-AP), ERK1/2 (11257-1-AP), and GAPDH (60004-1-lg) were purchased from Proteintech (Chicago, USA).

### MTT assay

MDA-MB-231 cells in logarithmic growth phase were seeded in 96-well plates ($4 \times 10^3$ cells/ well) and exposed to TRX at 0, 40, 80, and 120 μmol/L(μM). After culturing for 24, 48, and 72h, 20 μL of MTT solution (5 mg/ml) was added to each well and cultured for 4 h. After removing the supernatant solution, 110 μL of DMSO was added and vibrated to dissolve the formazan crystals. The optical density (OD) value was detected by a multifunctional microplate reader at 490 nm. The cell number (% of control): OD value (experimental group) / OD value (control group) * 100%.

### Wound healing assay

MDA-MB-231 cells in logarithmic growth phase were seeded in 12-well plates ($4 \times 10^5$ cells/ well). When the cells had adhered and reached 90% confluence, a 20 μL tip was used to scratch the bottom of the plate vertically. Then, the plates were washed twice with PBS after the medium was removed, and the cells were cultured in serum-free medium with 0, 40, or 80 μM TRX. The wounds were photographed by microscopy (OLYMPUS CKX53, 10x) after culture for 0, 24, and 48h, and the wound areas were measured by ImageJ software.

### Transwell assay

MDA-MB-231 cells in logarithmic growth phase were resuspended in serum-free medium and seeded in transwell inserts (24-well, 8.0 μM) with 200 μL serum-free medium ($8 \times 10^4$ cells/ insert). Then, 600 μL of medium with 10% FBS was added to the lower chambers. TRX

solution was added to final concentrations of the transwell inserts and lower chambers of 0 and 80 μM. After culture for 48 h, 4% paraformaldehyde and 0.1% crystal violet were added successively to the insert to fix and stain the cells. Five images of different visual fields for each insert were taken randomly under the microscope, and the migrated or invaded cells were counted. For invasion rather than migration assays, it is necessary to dilute the premelted Matrigel to 0.3 mg/ml with serum-free medium. Then, 100 μL of diluted Matrigel was added to the Transwell insert and incubated for 4 h. Cells could not be seeded until the Matrigel became solid.

## Western blot

MDA-MB-231 cells in logarithmic growth phase were seeded in 6-well plates ($4 \times 10^5$ cells/ well) and exposed to TRX at 0, 40, and 80 μM for 48 h. Total cellular proteins were extracted by RIPA buffer with protease inhibitor, and the concentrations were determined by the BCA protein concentration quantitative method. All samples were separated by 10% SDS-PAGE gel and transferred to PVDF membranes. The membranes were blocked in 5% skim milk at room temperature for 1 h, incubated in primary antibodies at 4°C overnight, washed three times with TBST solution for 10 min each and then incubated in secondary antibodies at room temperature for 1.5 h. The protein bands were developed by using hypersensitive enhanced chemi-luminescence (ECL), and the gray values were analyzed by ImageJ software.

## Network pharmacology

The 2-dimensional (2D) chemical structure of TRX was obtained from the PubChem database (https://pubchem.ncbi.nlm.nih.gov/) and then imported into SwissTargetPrediction [9] to predict targets, and those with a "probability>0.1" were selected as TRX targets. TNBC targets were obtained from GeneCards(https://www.genecards.org/), DrugBank [10], Therapeutic Target Database (TTD) [11] and OMIM (https://www.omim.org/) after removing duplicates. TRX targets and TNBC targets were imported into the Bioinformatics online website (http://www.bioinformatics.com.cn/) to obtain common targets and a Venn diagram. These common targets were imported into the STRING database (https://www.string-db.org/) to obtain the protein-protein interaction (PPI) network with the options "the high confidence>0.4" and "hide the disconnected nodes of the network". Meanwhile, GO and KEGG enrichment analyses were carried out in the DAVID database [12] with common targets, and the results were visualized through Micro Bioinformatics, an online bioinformatics analysis, visualization cloud platform.

## Molecular docking

The crystal structure of the candidate target as the receptor was obtained from the Research Collaboratory for Structural Bioinformatics Protein Data Bank (https://www.rcsb.org/) and imported into open-source PyMOL and AutoDockTools 1.5.7 software for pretreatment, which consisted of removing ligands, removing water and adding hydrogen bonds. Meanwhile, the 3D structure of TRX as the ligand was obtained from the PubChem database, converted into format by OpenBabel 2.4.1 software, and then imported into AutoDockTools for pretreatment, which included adding charges and limiting the ligand conformation. The processed receptor and ligand were subjected to molecular docking by AutoDock Vina. The results of molecular docking were imported into the Protein-Ligand Interaction Profiler (PLIP) web tool to obtain noncovalent interaction information and visualize it in PyMol.

### tBHQ experiment

MDA-MB-231 cells were divided into control, TRX, tBHQ, and TRX+tBHQ groups and treated with medium, 80 μM TRX, 10 μM tBHQ, 80 μM TRX and 10 μM tBHQ, respectively. Then, the cell migration and invasion abilities were detected by wound healing and Transwell assays, and the relative protein expression levels were detected by Western blot.

### Statistical analysis

All data statistical analysis and charts were provided by GraphPad Prism software, and the results are presented as the mean and standard deviation (mean ± SD). Student's t test was used to compare differences between two groups, and one-way ANOVA was used to compare multiple groups. A P value<0.05 was considered statistically significant.

## Results

### TRX inhibited the migration and invasion of MDA-MB-231 cells

To investigate the effect of TRX on MDA-MB-231 cells, we treated cells with different concentrations of TRX. The results of the MTT assay showed that TRX inhibited cellular viability, and the effect of 80 μM TRX was slightly obvious (Fig 1A). Compared to control area, the wound area of 0mM, 40mM and 80mM were 69.6%, 83.2% and 89.7% at 24h, the wound area of 0mM, 40mM and 80mM were 50.4%, 70.5% and 83.8% at 48h. The results of the wound healing assay showed that the migration of MDA-MB-231 cells was inhibited by TRX in a time- and concentration-dependent manner (Fig 1B and 1C). According to the Transwell assay (Fig 1D and 1E), we found that cells migration and invasion were decreased significantly after treatment with 80 μM TRX for 48 h. The inhibition rate of cell migration was 35.3%, and the inhibition rate of cell invasion was 46.3%. Epithelial mesenchymal transformation (EMT) is the key process of cancer cell metastasis, and cell

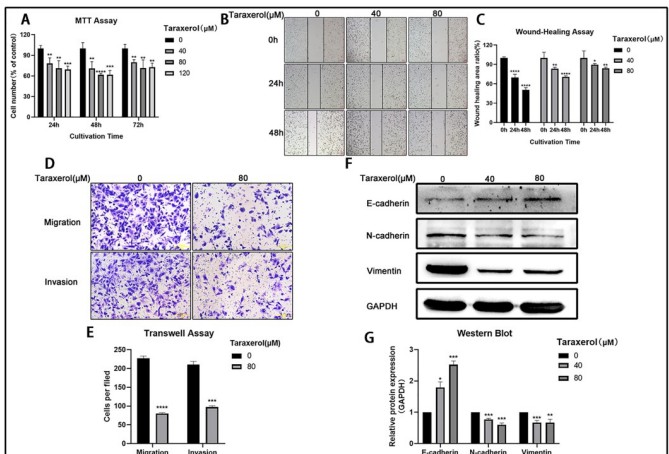

**Fig 1. TRX inhibited the migration and invasion of MDA-MB-231 cells.** A) Cell viability was assessed by MTT assay, and cell numberwere counted by GraphPad software 5.0. B) Cell migration was evaluated by wound healing assay. C) Statistical analysis of the wound healing area ratio. D) Cell migration and invasion were evaluated by Transwell assay. E) Statistical analysis of cells per field. F) The expression of EMT-related proteins was determined by Western blot. G) Statistical analysis of the relative protein expression. These results are representative of at least 3 independent experiments. Data are presented as the mean ± SD.*p<0.05, **p<0.01, ***p<0.001, ****p<0.0001 compared with the 0 μM group.

migration and invasion are greatly enhanced when epithelial cells acquire the characteristics of mesenchymal cells. Thus, we observed the expression of EMT markers by Western blot. The expression of the epithelial cell marker E-cadherin was upregulated, and the expression of the mesenchymal cell marker N-cadherin and Vimentin was downregulated (Fig 1F and 1G), suggesting that the migration and invasion of MDA-MB-231 cells were suppressed. These results indicated that TRX inhibited the migration and invasion of MDA-MB-231 cells.

## Prediction that MAPK3 was the critical target by network pharmacology and molecular docking

Fifty-four TRX targets with a probability>0.1 were predicted by importing the 2D chemical structure of TRX (Fig 2A), and 1751 TNBC targets were predicted from the GeneCards, DrugBank, TTD and OMIM databases. Then, we obtained 20 common targets by intersecting TRX targets and TNBC targets (Fig 2B), suggesting that more than 1/3 of TRX targets are closely related to TNBC. The PPI network (Fig 2C) was constructed through the STRING website and Cytoscape software with 20 common targets, in which the size and color depth of nodes had a direct correlation with the importance of targets. Thus, we considered MAPK3 to be the most promising candidate target. GO enrichment analysis included biological process, cellular component, and molecular function, and the results with P<0.01 are shown in Fig 2D. Notably, many biological processes and molecular functions were related to tyrosine phosphorylation, which was just right one of major phosphorylation forms of ERK1, a protein that was encoded by MAPK3.

There were 15 KEGG pathways (Fig 2E) enriched from common targets which could be divided three categories: cellular processes, organic systems and human diseases. Adherens junctions were the only cellular process, and the loss of cell adhesion was the first and most important step in cancer infiltration and metastasis, which was consistent with the results of the cell migration and invasion experiments. Furthermore, we could easily observe that MAPK3 was the critical target by drawing a Sankey bubble diagram (Fig 2F) to visualize the common target and results of KEGG enrichment analysis. According to molecular docking (Fig 2G), we found that MAPK3 could bind to TRX stably, and the noncovalent interactions between them included hydrogen bonds, hydrophobic interactions and salt bridges (Table 1). Therefore, we believe that MAPK3 is a critical target by which TRX inhibits the migration and invasion of MDA-MB-231 cells.

## TRX inhibited the migration and invasion of MDA-MB-231 cells via the ERK/Slug axis

Adherens junction was the only pathway closely related to cell migration and invasion, which ranked high in the KEGG enrichment analysis results. Meanwhile, in this pathway, we found that ERK1 could affect E-cadherin by acting on Slug (Fig 3A). Thus, we speculated that TRX might inhibit the migration and invasion of MDA-MB-231 cells via the ERK/Slug axis. Then, we detected the protein expression levels of p-ERK, ERK and Slug in MDA-MB-231 cells after TRX treatment. The Western blot results showed that TRX significantly suppressed the expression of p-ERK and Slug in a time- and concentration-dependent manner. The total protein expression level of ERK showed no obvious change (Fig 3B and 3C). These results indicated that TRX could inhibit the migration and invasion of MDA-MB-231 cells via the ERK/Slug axis.

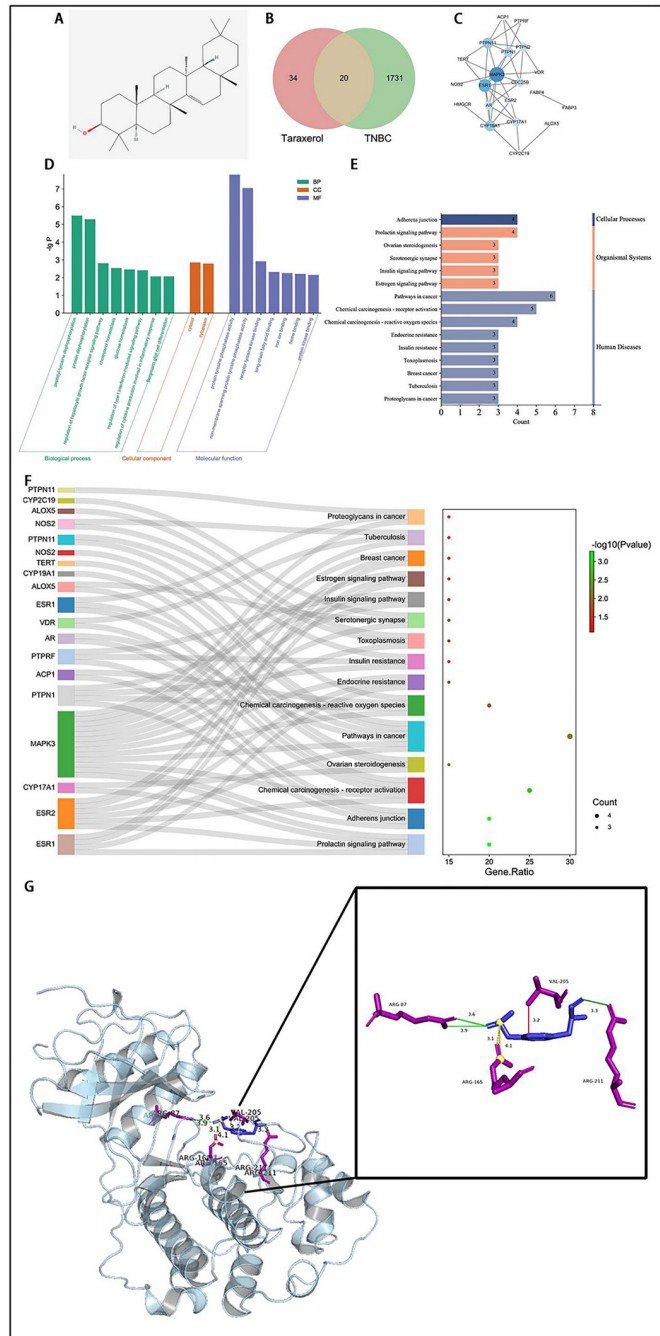

**Fig 2. Prediction that MAPK3 was the critical target by network pharmacology and molecular docking.** A) The 2D chemical structure of TRX. B) Venn diagram of TRX and TNBC targets. C) The PPI network of 20 common targets. D), E) Histogram of GO and KEGG enrichment analysis results. F) The Sankey bubble diagram with common targets and KEGG enrichment analysis results. G) Molecular docking of TRX onto the MAPK3 protein. Molecules are depicted by a stick model, the HBonds, Hydrophobic and salt bridges are depicted by green, red and yellow lines, respectively, and the distances are shown in angstroms.

**Table 1. Molecular docking for MAPK3 and TRX.**

| Target | PDB ID | Affinity | Interactions | | |
|---|---|---|---|---|---|
| | | (Kcal/mol) | Type | Residue | Distance |
| MAPK3 | 2ZOQ | -9.4 | hydrogen bonds | ARG87, ARG211 | 3.6, 3.9, 3.3 |
| | | | hydrophobic interactions | VAL205 | 3.2 |
| | | | salt bridges | ARG165 | 4.1, 3.1 |

## ERK activator(tBHQ) reversed the TRX-induced suppression of MDA-MB-231 cell migration and invasion

To further confirm that TRX inhibits the migration and invasion of MDA-MB-231 cells via the ERK/slug axis, we observed whether tBHQ could reverse the TRX-induced suppression of MDA-MB-231 cell migration and invasion by wound healing and Transwell assays. In wound-healing assay, compared to control group, the he wound area of TRX group were 91.1% at 24h, 89.6% at 48h. While the wound area of TRX+tBHQ group were 85.6% at 24h, 73.5% at 48h. In the cell migration of transwell assay, the cell number of TRX group was 230 per filed, while the TRX+tBHQ group was 356 per filed. In the cell invasion of transwell assay, the cell number of

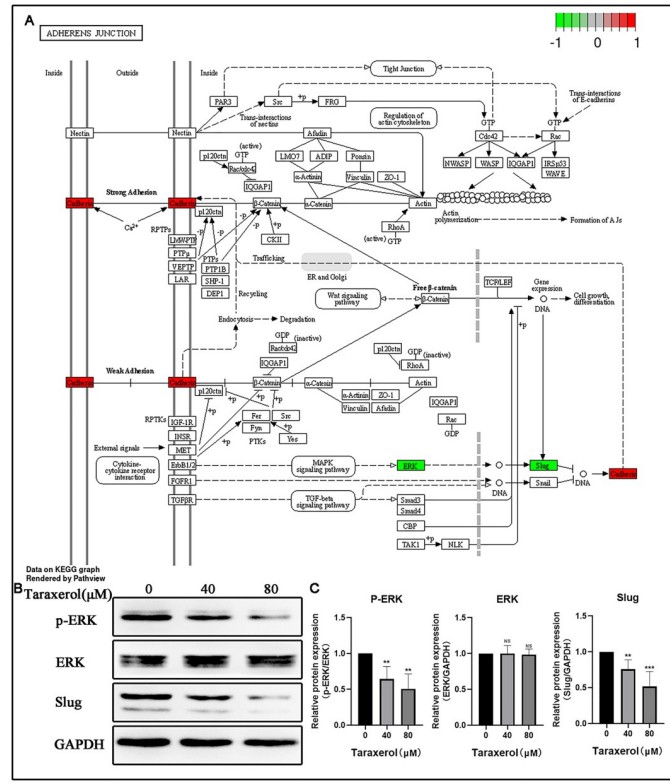

**Fig 3. TRX inhibited the migration and invasion of MDA-MB-231 cells via the ERK/Slug axis.** A) Map04520, named the adherens junction, shows that the ERK/Slug axis might be the mechanism. The red of scale indicates a positive correlation between gene and TRX, while the green indicates a negative correlation. B) The expression of p-ERK, ERK and Slug was determined by Western blot. C) Statistical analysis of protein expression in MDA-MB-231 cells. These results are representative of at least 3 independent experiments. Data are presented as the mean ± SD. NS, no significant difference, **p<0.01, ***p<0.001 compared with the 0 µM group.

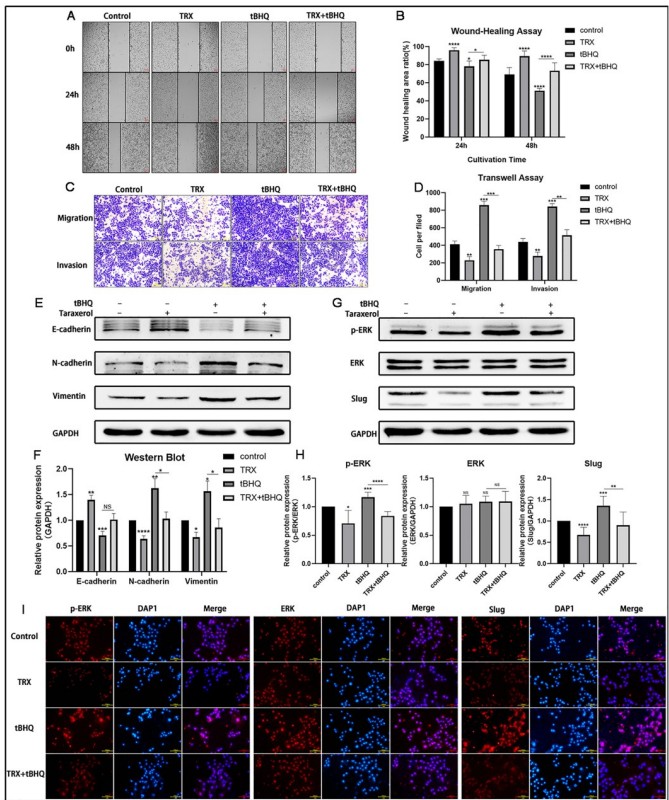

**Fig 4. ERK activator (tBHQ) reversed the TRX-induced suppression of MDA-MB-231 cell migration and invasion.** A) Cell migration was evaluated by wound healing assay. B) Statistical analysis of the wound healing area ratio. C) Cell migration and invasion were evaluated by Transwell assay. D) Statistical analysis of the cells per field. E), G) The expression of relative proteins was determined by Western blot. F), H) Statistical analysis of the relative protein expression. I) Immunofluorescence images of p-ERK, ERK and Slug (red) expression following control, TRX, tBHQ, TRX+tBHQ treatment. DAP1 staining was done to visualize nuclei (blue). These results were representative of at least independent experiments. Data are presented as the mean ± SD. NS, no significant difference, *p<0.05, **p<0.01, ***p<0.001, ****p<0.0001 compared with the control group.

TRX group was 279 per filed, while the TRX+tBHQ group was 515 per filed. These results indicated that tBHQ could partially reverse the TRX-induced suppression of MDA-MB-231 cell migration and invasion (Fig 4A–4D). Moreover, we detected the expression of EMT-related proteins (E-cadherin, N-cadherin, Vimentin) and axis-related proteins (p-ERK, ERK, Slug) by Western blot and observed that tBHQ could partially reverse the increase in the expression of E-cadherin and the decrease in N-cadherin, Vimentin, Slug and ERK phosphorylation caused by TRX (Fig 4E–4H). Moreover, we detected axis-related proteins with immunofluorescence. The results shown in Fig 4I indicated that TRX could decrease the expression of ERK phosphorylation and Slug. In this case, we believed that TRX could indeed inhibit the migration and invasion of MDA-MB-231 cells via the ERK/slug axis.

## Discussion

According to the latest data published by the International Agency for Research on Cancer (IARC) [13], BC has become the most common cancer worldwide. Improving the awareness and ability of BC prevention and treatment is one of the most important measures to ensure women's health. Although, overall, BC has a better survival than others, postoperative patients

with TNBC usually have no appropriate medicine options due to the lack of sufficient receptor expression. Therefore, it is necessary to promote research innovation drugs.

It is well known that the participation of TCM could enhance efficacy and reduce toxicity in a great measure during the process of tumor treatment [14]. Taraxacum, as a pure natural herb, is a heat-clearing TCM. The broad application of different taraxacum extracts has shown that they could inhibit the proliferation and metastasis of TNBC [15], and TRX has been proven to have an inhibitory effect on a variety of tumor cell lines. In this study, we found that the migration and invasion of MDA-MB-231 cells were inhibited by TRX through wound healing and Transwell assays.

EMT, a process in which epithelial cells acquire the characteristics of mesenchymal cells, is considered to be closely related to cell migration and invasion, causing cells to become more invasive with the loss of polarity and adhesive ability [16]. E-cadherin, N-cadherin and Vimentin are classic markers of EMT. The expression level of E-cadherin was negatively correlated with BC cell migration and invasion [17], and in dedifferentiated breast cancer cells, it was significantly lower than that in differentiated cells [18]. N-cadherin with high expression in invasive BC cells could promote cell metastasis by interacting with surrounding stromal cells [19]. Vimentin is highly expressed in invasive BC cells and enhances cell metastasis by promoting the EMT process [20]. Therefore, we detected the protein expression levels of E-cadherin, N-cadherin and Vimentin by Western blot. After TRX treatment, E-cadherin expression was upregulated, while N-cadherin and Vimentin expression was downregulated, indicating that the EMT process in MDA-MB-231 cells was inhibited. Combined with the previous results, we believed that TRX could inhibit the migration and invasion of MDA-MB-231 cells.

Network pharmacology has been widely used in TCM research because it has the same integrity and systematic characteristics as TCM. Based on the theoretical basis of systems biology, it combines bioinformatics with network analysis to study the mechanism of medicine effects at the system level. Meanwhile, molecular docking, as a commonly used method for drug screening, can directly reveal the interactions between drugs and targets and predict their affinity. In this study, we observed that MAPK3 was the critical target by which TRX inhibited the migration and invasion of MDA-MB-231 cells by network pharmacology, and it could stably bind to TRX by molecular docking. Extracellular regulated protein kinase 1 (ERK1), encoded by MAPK3, belongs to the mitogen-activated protein kinase (MAPK) family [21]. Extracellular regulated protein kinase 2 (ERK2), encoded by MAPK1 is highly similar to ERK1 in sequence, protein function, upstream activation pathways and downstream targets. Therefore, ERK1 and ERK2 are often collectively referred to as ERK [22]. ERK is usually located in the nucleus and is transferred to the cytoplasm after being phosphorylated under various stress states [23]. ERK is basically activated by dual phosphorylation at threonine and tyrosine sites [24,25], hence, the protein tyrosine phosphatase activity predicted by GO enrichment analysis is necessary for ERK activation [26]. Previous studies have shown that the phosphorylation level of ERK is closely related to tumor occurrence and development. The inhibition of ERK phosphorylation could suppress cancer-stromal interactions in pancreatic cancer [27] and promote the ULK1 degradation process, leading to an improved invasive phenotype under hypoxia and osteolytic bone metastasis in BC cells induced by ULK1 deficiency [28]. Otherwise, among the 15 pathways obtained from KEGG enrichment analysis, adherens junction was the most significant pathway and closely related to cell migration and invasion. ERK is located in this pathway and acts on Slug to regulate cell adhesion, leading to cadherins switching. The Western blot results showed that TRX significantly reduced the expression levels of ERK phosphorylation and Slug. Therefore, we hypothesized that TRX could inhibit the migration and invasion of MDA-MB-231 cells via the ERK/Slug axis and verified this hypothesis by adding an ERK activator.

However, we acknowledge that there are still several limitations of this study. First, we simply detected the expression changes of EMT markers but did not show the EMT in more detail by cellular immunofluorescence staining. Second, further experiments in vivo are essential to verify the molecular mechanism. Moreover, due to the high invasiveness of TNBC, MDA-MB-231 cells were selected for this study. However, the other three subtypes of BC should also be studied to determine the therapeutic effect of TRX on BC. MTT assay relies on cell number to determine cell viability, while the experimental results are greatly influenced by the seeding density. Thus, it is necessary to support this conclusion through other experiments.

## Conclusion

In summary, we illustrated that TRX could inhibit the migration and invasion of MDA-MB-231 cells, and the mechanism by which TRX inhibited the migration and invasion of MDA-MB-231 cells via the ERK/Slug axis was elucidated by network pharmacology, molecular docking and Western blot. Thus, TRX may be a promising therapeutic strategy for blocking tumorigenesis in TNBC.

## Supporting information

**S1 Raw images.**
(ZIP)

## Author Contributions

**Conceptualization:** Lu Chen, Zhi-hua Li.

**Data curation:** Yu-ting Xia, Yu-qin Zhang, Lu Chen.

**Formal analysis:** Yu-ting Xia, Lu Chen.

**Funding acquisition:** Yu-ting Xia, Zhi-hua Li.

**Investigation:** Yu-ting Xia, Da Huang, Yulu Zhang, Cong Li.

**Methodology:** Yu-ting Xia, Yu-qin Zhang, Lu Chen, Liangliang Min, Da Huang, Yulu Zhang, Cong Li.

**Project administration:** Yu-ting Xia, Zhi-hua Li.

**Resources:** Yu-ting Xia, Lu Chen.

**Software:** Liangliang Min.

**Supervision:** Zhi-hua Li.

**Validation:** Yu-ting Xia.

**Writing – original draft:** Yu-ting Xia, Lu Chen.

**Writing – review & editing:** Zhi-hua Li.

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
