## [Decision Letter · Decision Letter 0]

25 May 2023

PONE-D-23-09759Suppression of migration and invasion by taraxerol in the triple-negative breast cancer cell line MDA-MB-231 via the ERK/Slug axisPLOS ONE

Dear Dr. li,

Thank you for submitting your manuscript to PLOS ONE. After careful consideration, we feel that it has merit but does not fully meet PLOS ONE’s publication criteria as it currently stands. Therefore, we invite you to submit a revised version of the manuscript that addresses the points raised during the review process. The manuscript of Xia et al. is interesting. However, there are many flaws of the current version of the manuscript pointed out by the reviewers. Please address all the reviews and academic editor comments.    Academic Editor comments:  Please express MTT assay results as viable cells (% of control), change the figure 1a accordingly, and add information about limitations of MTT assay in the discussion part. You can find relevant information here - DOI: 10.1097/CAD.0000000000001131.Please perform molecular dynamic simulation to validate docking results. Manuscript needs significant English improvement.

We look forward to receiving your revised manuscript.

Kind regards,

Salman Shakil

Academic Editor

PLOS ONE

Journal Requirements:

"This work was supported by Natural Science Fund in Jiangxi Province (Contract grant number: 20202BAB206046) and Jiangxi Provincial Postgraduate Innovation Special Fund Project (Contract grant number: YC2021-S500)."

6. Please upload a new copy of Figure 2D as the detail is not clear. Please follow the link for more information: " ext-link-type="uri" xlink:type="simple">https://blogs.plos.org/plos/2019/06/looking-good-tips-for-creating-your-plos-figures-graphics/"
" ext-link-type="uri" xlink:type="simple">https://blogs.plos.org/plos/2019/06/looking-good-tips-for-creating-your-plos-figures-graphics/"

Reviewers' comments:

Reviewer's Responses to Questions

**Comments to the Author**

1. Is the manuscript technically sound, and do the data support the conclusions?

Reviewer #1: Partly

Reviewer #2: Partly

Reviewer #3: Partly

2. Has the statistical analysis been performed appropriately and rigorously? 

Reviewer #1: Yes

Reviewer #2: Yes

Reviewer #3: I Don't Know

3. Have the authors made all data underlying the findings in their manuscript fully available?

Reviewer #1: Yes

Reviewer #2: Yes

Reviewer #3: Yes

4. Is the manuscript presented in an intelligible fashion and written in standard English?

Reviewer #1: Yes

Reviewer #2: Yes

Reviewer #3: No

5. Review Comments to the Author

Reviewer #1: This article "Suppression of migration and invasion by taraxerol in the triple-negative breast cancer cell line MDA-MB-231 via the ERK/Slug axis" is interesting, but I feel it's not a completed work. Authors focused on Slug only but there are important transcription factors (e.g., Twist, Snail, Zeb etc), especially Twist1 is a master regulator of cadherins as well as a key factor in EMT associated metastasis. What's the role of taraxerol on Twist in MDA-MB-231? Also, author claims that taraxerol suppress cell migration and invasion by ERK/Slug axis. Is it direct suppression or indirect? May be, it is the secondary effect of other mechanism? May be reporter assay could clarify this. To consider this paper for publishing, I think these issues need to be clarified.

Reviewer #2: The manuscript is intriguing as it explores the mechanistic aspects of taraxerol (TRX)-induced anti-migratory and anti-invasive effects on MDA-MB-231 cells. The study reveals that TRX inhibits the migration and invasion of MDA-MB-231 cells by targeting the ERK/Slug pathway. Nonetheless, prior to considering publication in PLOS ONE, there are several noteworthy issues that need to be addressed.

Comments:

1. The manuscript is lacking specific information about MDA-MB-231 cells. It is recommended to provide details about MDA-MB-231 cells in their initial mention within the introduction of the manuscript.

2. In Figure 1b, the cell density at 0 hours appears inconsistent across all the wells. Please observe the 80 µM TRX-treated well at 0 hours and the 0 µM TRX-treated treated well at 0 hours, as clear disparities in cell concentration are evident. Are there any representative photographs available that show equal cell density in all the wells at the initial time point?

3. The authors' bioinformatics approach in identifying MAPK3 as the target of TRX is quite fascinating. However, figure 2 is not acceptable to me because of its poor resolution. I was unable to verify the details in the figure due to its blurry appearance. Kindly submit a higher-resolution version of Figure 2 for review.

4. Once again, Figure 3a is deemed unacceptable due to its blurry appearance. Kindly provide an improved figure with a higher resolution.

5. In the text of the Results section, it is mentioned that the Western blot results indicate a significant suppression of p-ERK and Slug expression by TRX in a time- and concentration-dependent manner. However, upon examining Figure 3c, it is apparent that the authors only presented a concentration-dependent effect of TRX, and no demonstration of a time-dependent effect is evident.

6. The mechanistic details of TRX-induced anti-migratory and anti-invasive effects on MDA-MB-231 cells have been supported by Western blotting data. However, to enhance the strength of the findings and improve the manuscript's quality, the authors could consider including immunofluorescence staining or any other supplementary data. Incorporating imaging data would provide additional support and bolster the validity of their conclusions. It is recommended that the authors consider this suggestion.

Reviewer #3: The manuscript by Xia et al. describes the role of Taraxerol (TRX) in inhibiting the migration and invasion of breast cancer cell line, MDA-MB-231. They identified a possible in-vitro mechanism, ERK/SLUG axis, by which TRX executed its anti-tumour function in this cell line. They concluded that TRX can be a potential therapeutic target that acts by restricting the tumorigenesis during Breast cancer. The subject of the study is interesting and timely. The experimental design in this manuscript is sound, and the conclusions are important.

However, I have some concerns. The manuscripts are not written in a proper English standard. Some of the statements are vague and the flow of the sentence is very often interrupted. Also, authors need to be consistent in their way of writing. I would recommend reviewing the manuscript by professional english editors before it gets accepted for publication. Additionally, I would need to see more data and rigorous support for their claims before I can recommend that this manuscript be published.

Introduction:

1. No space between the end of sentence and the references in the introduction section. This has been observed throughout the manuscript.

2. Authors need to elaborate the meaning of TNBC and TRX before they start using it in the sentence.

Materials and methods:

1. Authors need to mention what kind of cell line MDA-MB-231 is?

2. MTT assay: The cell count should be written as 4 x 103 cells.

3. Wound healing assay: Mention the model of microscope and what kind of microscope was used to capture images.

4. Authors need to mention the version of any bioinformatics or statistical software used throughout the manuscript. For example: GraphPad Prism, ImajeJ software etc.

5. Provide catalogue details for reagents such as Matrigel.

6. Western blotting: Authors need to rewrite the last sentence as the photographs cannot be taken by ECL rather the bands were developed using ECL reagents.

7. Network Pharmacology: Why the cut-off of probability0.1 was selected for TRX target?

8. Network pharmacology: Authors need to give a clear explanation about how the results were visualized by bioinformatics in the last sentence.

Results:

In the result section, my major concern is authors need to explain the results before/after giving any interpretation which was missing throughout the manuscript. Also, Authors need to mention how much increase/decrease they have noticed in numbers throughout the result sections.

With all the Western blots figure, the author needs to mention if they have stripped and re-probed the blot. If so, they need to describe the stripping procedure in the method section.

1. Cell viability: Authors should plot the graphs as cell viability (%) since OD value hardly makes any sense here.

3. Figure 1C: Authors need to have one more set of bars with 0h at 0 mm, 40 mM and 80 mM. The statistical analysis for the treated groups should be based on this set.

4. Figure 1C: Authors need to be elaborative in writing down the percentage results in the result section. When it says, decreased/ increased significantly readers need to know to which extent the differences were observed.

5. Figure 1D, 1E: No proper explanation of the results. Authors need to mention clearly which reading (Migration or invasion) was taken from which (upper and lower) well?

6. Figure 2: What are these 20 targets? Include a table as a supplementary file.

7. Figure 2: As authors have mentioned MAPK3 as the most promising candidate, I would like to see the protein expression level of MAPK3 alongside ERK/p-ERK by western blotting in Figure 3B/4G. Also, if authors are specifically saying that MAPK3 is involved then why they are measuring total p-ERK/ERKs instead of p-ERK1/ERK1? This is contradictory as MAPK1 can also be a major candidate here based on the observation of the western blots in figure 3B and 4G.

8. Fig 3A: Did author measure the expression of Cadherin in the blots? If not, why the Cadherin in this figure is marked as red? There is no explanation about cadherin in this section. They also need to explain clearly in the figure legend about the scales drawn on the top right side.

9. Fig 3B,C: Did authors measure the expression of p-ERK1/ERK1 or both ERKs? If the probing for p-ERK, ERK and GAPDH was done on the same blot after stripping, how does the author confirm that p-ERK and ERK does not cross-react? Authors need to describe how did they quantify p-ERK and ERK level?

10. Fig 3B,C: “TRX significantly suppressed the expression of p-ERK and Slug in a time- and concentration-dependent manner”: Wrong statement as authors did not measure the time-dependent effect here.

11. Fig 4A,D: Authors need to give proper explanation for their interpretation?

Discussion:

1. Need reference for this statement: According to the latest data published by the International Agency for Research….

2. This sentence doesn’t make sense: Therefore, it is necessary to promote research no innovation drugs..

3. Is it dedifferentiated? : and in dedifferentiated breast cancer cells, it was significantly..

Figure legend:

All the legends need to be re-written by giving more scientific details of the experiments. Authors need to mention what statistical analysis they have used for each of the experiment. Is it one/two-way ANOVA or t-test?

Figure: 1A: Wrong statement as the OD values cannot be counted by GraphPad software

6. PLOS authors have the option to publish the peer review history of their article (what does this mean?). If published, this will include your full peer review and any attached files.

Reviewer #1: No

Reviewer #2: No

Reviewer #3: **Yes: **Farjana Ahmed

---

## [Author Response · Author response to Decision Letter 0]

9 Jul 2023

Dear Editor:

Thank you for your and the reviewers’ comments on our manuscript entitled “Suppression of migration and invasion by taraxerol in the triple-negative breast cancer cell line MDA-MB-231 via the ERK/Slug axis”. (EMID: f1702d110dff026b). Our deepest gratitude goes to the anonymous reviewers for their careful work and thoughtful suggestions that have helped improve this manuscript substantially. Meanwhile, thank you for giving us a chance to revise and resubmit our manuscript.

Based on these comments, we have made careful modifications again. All changes made to the manuscript are marked in red. We modified the figures adhere fully to these guidelines and provide the original underlying images for all blot or gel data reported in Supporting Information . These changes will not influence the content and framework of the manuscript. We hope that these revisions are satisfactory and the revised manuscript will be acceptable for publication.

Below you will find our point-by-point responses to your comments/questions. Once again, thank you for your comments and suggestions.

Sincerely,

XIA Yu-ting, ZHANG Yu-qin, CHEN Lu, MIN Liang-liang, HUANG Da, ZHANG Yu-lu, LI Cong, LI Zhi-hua*

*Corresponding author. E-mail address: huazhili0802@163.com

---

## [Decision Letter · Decision Letter 1]

4 Sep 2023

Suppression of migration and invasion by taraxerol in the triple-negative breast cancer cell line MDA-MB-231 via the ERK/Slug axis

PONE-D-23-09759R1

Dear Dr. Li,

We’re pleased to inform you that your manuscript has been judged scientifically suitable for publication and will be formally accepted for publication once it meets all outstanding technical requirements.

Kind regards,

Salman Shakil

Academic Editor

PLOS ONE

Additional Editor Comments (optional):

Reviewers' comments:

Reviewer's Responses to Questions

**Comments to the Author**

1. If the authors have adequately addressed your comments raised in a previous round of review and you feel that this manuscript is now acceptable for publication, you may indicate that here to bypass the “Comments to the Author” section, enter your conflict of interest statement in the “Confidential to Editor” section, and submit your "Accept" recommendation.

Reviewer #1: (No Response)

Reviewer #2: All comments have been addressed

2. Is the manuscript technically sound, and do the data support the conclusions?

Reviewer #1: (No Response)

Reviewer #2: Yes

3. Has the statistical analysis been performed appropriately and rigorously? 

Reviewer #1: (No Response)

Reviewer #2: Yes

4. Have the authors made all data underlying the findings in their manuscript fully available?

Reviewer #1: (No Response)

Reviewer #2: Yes

5. Is the manuscript presented in an intelligible fashion and written in standard English?

Reviewer #1: (No Response)

Reviewer #2: Yes

6. Review Comments to the Author

Reviewer #1: No more comments on "Suppression of migration and invasion by taraxerol in the triple-negative breast cancer cell line MDA-MB-231 via the ERK/Slug axis"

Reviewer #2: All the comments, suggestions, and queries have been convincingly addressed by the authors. I recommend accepting the article for publication in PLOS ONE.

7. PLOS authors have the option to publish the peer review history of their article (what does this mean?). If published, this will include your full peer review and any attached files.

Reviewer #1: No

Reviewer #2: No

---

## [Editor Report · Acceptance letter]

18 Sep 2023

PONE-D-23-09759R1 

Suppression of migration and invasion by taraxerol in the triple-negative breast cancer cell line MDA-MB-231 via the ERK/Slug axis 

Dear Dr. Li:

I'm pleased to inform you that your manuscript has been deemed suitable for publication in PLOS ONE. Congratulations! Your manuscript is now with our production department. 

Kind regards, 

on behalf of

Dr Salman Shakil 

Academic Editor

PLOS ONE